# Mating-Induced Common and Sex-Specific Behavioral, Transcriptional Changes in the Moth Fall Armyworm (*Spodoptera frugiperda*, Noctuidae, Lepidoptera) in Laboratory

**DOI:** 10.3390/insects14020209

**Published:** 2023-02-19

**Authors:** Ting Wu, Da-Hu Cao, Yu Liu, Hong Yu, Da-Ying Fu, Hui Ye, Jin Xu

**Affiliations:** 1Yunnan Academy of Biodiversity, Southwest Forestry University, Kunming 650224, China; 2Yunnan Key Laboratory of Plateau Wetland Conservation, Restoration and Ecological Services, Southwest Forestry University, Kunming 650224, China; 3School of Life Science, Southwest Forestry University, Kunming 650224, China; 4School of Ecology and Environment, Yunnan University, Kunming 650091, China

**Keywords:** *Spodoptera frugiperda*, reproductive behavior, fecundity, RNAseq, mating-induced transcriptional changes

## Abstract

**Simple Summary:**

In this study, we found mating-induced sex-specific behavioral and transcriptional changes, and the transcriptional variation is consistent with postmating physiological and behavioral alteration in each sex. Virgin females and males showed high calling or courting behavior, whereas mated females and males showed very low calling or courting behavior. Mated females started to lay eggs at the beginning of the next night after the first mating. Obvious differences on female oviposition patterns were found between different mate treatments, suggesting that females may use a fertilization or oviposition strategy to obtain indirect genetic benefits. Differential expression analysis indicated that mating induced upregulation of many reproductive related genes and soma maintenance related genes in females. Mating in males also induced upregulation on soma maintenance related genes immediately after mating (0 h postmating), but induced downregulation on these genes after a period of time (6–24 h postmating).

**Abstract:**

The intermediate process between mating and postmating behavioral changes in insects is still poorly known. Here, we studied mating-induced common and sex-specific behavioral and transcriptional changes in both sexes of *Spodoptera frugiperda* and tested whether the transcriptional changes are linked to postmating behavioral changes in each sex. A behavioral study showed that mating caused a temporary suppression of female calling and male courting behavior, and females did not lay eggs until the next day after the first mating. The significant differences on daily fecundity under the presence of males or not, and the same or novel males, suggest that females may intentionally retain eggs to be fertilized by novel males or to be fertilized competitively by different males. RNA sequencing in females revealed that there are more reproduction related GO (gene ontology) terms and KEGG (Kyoto encyclopedia of genes and genomes) pathways (mainly related to egg and zygote development) enriched to upregulated DEGs (differentially expressed genes) than to downregulated DEGs at 0 and 24 h postmating. In males, however, mating induced DEGs did not enrich any reproduction related terms/pathways, which may be because male reproductive bioinformatics is relatively limited in moths. Mating also induced upregulation on soma maintenance (such as immune activity and stress reaction) related processes in females at 0, 6 and 24 h postmating. In males, mating also induced upregulation on soma maintenance related processes at 0 h postmating, but induced downregulation on these processes at 6 and 24 h postmating. In conclusion, this study demonstrated that mating induced sex-specific postmating behavioral and transcriptional changes in both sexes of *S. frugiperda* and suggested that the transcriptional changes are correlated with postmating physiological and behavioral changes in each sex.

## 1. Introduction

The fall armyworm, *Spodoptera frugiperda* (Lepidoptera: Noctuidae), is native to subtropical and tropical regions in the Americas. This pest now has two strains: the corn strain and rice strain [1]. Its caterpillars mainly attack corn and can cause substantial economic losses [2,3]. It is estimated that this pest can cause 21% to 53% of the loss of corn yield (worth $2481 m–$6187 m) every year in Africa [4]. This pest was first detected in Southwest China at the end of 2018, which then quickly spread to vast areas of China and caused huge losses [5,6,7]. The potential economic losses to corn caused by this pest every year in China range from $17,286 m to $52,143 m [8]. This moth pest is also notorious for its long-distance migration ability [9,10], strong insecticide resistance [11,12] and high reproductive fitness [13]. At present, broad-spectrum chemical pesticides are mainly used to control this pest, which further improves its resistance to insecticides [11,12,14]. Sustainable and environment-friendly control strategies are required for better control of this pest.

Insect reproductive behavior is an important basis of sexual selection and evolutionary theories, and also has important value in pest control [15,16]. Insect reproductive behavior mainly includes female calling, male courtship, mating, oviposition and other behavioral processes directly or indirectly related to reproduction [17]. Mating is an essential behavioral process of sexual reproduction and a turning point of significant changes in female physiology and behavior [17,18]. Before mating, female insects, in general, actively engage in calling behavior and release sex-pheromones, rarely lay eggs and easily accept male courtship. After mating, females rarely show calling behavior and do not easily accept male courtship and start to search hosts and lay a large number of eggs [18,19]. Molecular studies in *Drosophila* have identified a large number of male accessory gland proteins (Acps), such as Acp26Aa (Ovulin) that can promote ovulation, and Acp70A (Sex peptide, SP) that can affect female reproductive behavior [20]. A later study in *Drosophila* further isolated a SP receptor (SPR) [21]. Studies in some other insect species also found such functional Acps, such as Aea-HP-1 in *Aedes aegypti* [22]. However, the identification of Acps and their receptors, particularly functional evidence, is scarce or limited in many other insect species.

In Lepidoptera, several studies have shown that mating also results in marked changes in the behavior and physiology of adult moths, particularly in females [23,24,25,26]. Studies in noctuid moths, such as *Spodoptera litura* [27] and *Ephestia kuehniella* [28], demonstrated that mating induced remarkable changes in female behavior and physiology, and determined that factors from male accessory gland (MAG) secretions play a key role in such switches in females after mating, such as suppressed female sexual receptivity and stimulating egg development. Further studies also found the existence of SPR in noctuid moths, such as *Helicoverpa armigera* [29,30] and *S. litura* [31], and showed that this receptor functions in mediating female post-mating behavior. These results imply the existence of SP-like ligands in moths, but still lack direct evidence.

Therefore, the postmating changes in female physiology and behavior (results) caused by male mating factors (causes) are obvious, and the relationship between the results and causes has been well studied. However, the intermediate processes or linkages between the results and causes are still poorly known.

Studies in *Drosophila* [32,33,34,35] and other insect species [36,37,38] demonstrated that mating can cause significant transcriptional changes in females [32,33,34,35], which may support them to obtain a high rate of fertilization and fecundity. These studies on different insect species, based on depth sequencing and bioinformatics, have provided further insights in this field. In addition, some of these studies also showed that mating may trigger trade-offs between reproduction and soma maintenance (such as immunity, stress response, DNA/protein repair and longevity) [39,40]. For instance, a recent study in *S. litura*, based on female whole-body RNA sequencing (RNAseq) analysis, found that mating may induce a trade-off between immunity and reproduction [41]. However, some other studies in different insect species also found inconsistent results, such as mating-induced upregulation of immune-related transcripts in the female oviduct [42]. These studies suggest that mating-induced transcriptional regulation and possible trade-offs should be species specific. Moreover, previous studies on mating-induced changes and trade-offs mainly focused on females. Future studies in both sexes of different insect species will not only promote our understanding of the molecular machinery behind mating, but will also help to identify important gene targets that might be useful in the future management of pests.

In the present study, therefore, we studied mating-induced behavioral and transcriptional changes in *S. frugiperda* males and females and tested whether the transcriptional changes were related to postmating behavioral changes in each sex. We hypothesize that (1) mating induces sex-specific behavioral and transcriptional changes, (2) the transcriptional changes are consistent with postmating physiological and behavioral changes and (3) mating triggers trade-offs between reproduction and soma maintenance. To test these hypotheses, we studied the mating-induced common and sex-specific transcriptional and behavioral changes in *S. frugiperda*. We then discussed the evolutionary significance of the postmating gene expression regulation between reproduction and soma maintenance and the molecular machinery behind the postmating switches.

## 2. Materials and Methods

### 2.1. Insects

The larvae of *S. frugiperda* were collected in corn fields near Dongchuan town in Yunnan Province, China. The collected larvae were reared on an artificial diet that was prepared in the IPM Laboratory of Southwest Forestry University (Kunming, China), according to the recipe outlined by Li et al. [43], with some modifications. To prepare the diet, 100 g wheat bran, 100 g soybean meal, 40 g yeast powder, 30 g casein and 30 g agar powder were added to 1 L distilled water and mixed briefly using a spatula. The mixture was steamed for 1.5 h by using a steam pot and an induction cooker for heating at 1600 W. After the steamed mixture was cooled to 50 °C, 3 g vitamin C, 2 g cholesterol and 3 g potassium sorbate were added and mixed thoroughly using the spatula to obtain the diet. The diet was stored at 4 °C before use. The rearing condition was set as 28 ± 1 °C, 60–80% relative humidity and a 14:10 h (light:dark) photoperiod. To ensure virginity, male and female pupae were sexed based on morphological characteristics [44] and then caged separately. Newly enclosed male and female moths were collected and reared in different cages under the same conditions and fed a 10% honey solution.

### 2.2. Reproductive Behavior

Three day old virgin male and female moths (sexually mature) [45] were used for the study of behavioral differences before and after mating, under different mating conditions, for two nights.

In the first night, five treatments were set: (1) singly caged virgin female—a 3-d-old virgin female was caged (n = 20) and female calling and oviposition behavior was observed; (2) paired—a 3-d-old virgin female was caged with a 3-d-old virgin male (n = 40) and male courtship, female calling, mating and oviposition were recorded; (3) change male mate—a 3-d-old virgin female was caged with a 3-d-old virgin male (male 1#) (n = 40). If mating happened, the mated male was removed immediately after mating and a novel 3-d-old virgin male (male 2#) was introduced. Female calling, mating and oviposition, and the courting of male 2#, were recorded; (4) change female mate—a 3-d-old virgin male was caged with a 3-d-old virgin female (female 1#) (n = 40). If mating happened, the mated female was removed immediately after mating and a novel 3-d-old virgin female (female 2#) was introduced. Male courtship and mating events, and the calling of female 2#, were recorded; (5) one-time mated female—a 3-d-old virgin female was caged with a 3-d-old virgin male (n = 40). If mating happened, the male was removed immediately after mating. Female calling and mating events were recorded. Please see below for detailed observation and recording methods.

About 57% of paired moths [treatments (2)–(5)] mated one time in the first night (see results); thus, in the second night, all the 20 singly caged virgin females, as well as the 20 mated pairs from paired, 20 mated females with their novel mates from change male mate, 20 mated males with their novel mates from change female mate and 20 mated females from one-time mated female were randomly selected, and their reproductive behavior was observed and recorded, as below.

Test insects were caged in plastic boxes (25 × 15 × 8 cm) with one moth or one pair per box. Each box was placed with a petri dish containing a cotton and honey solution, and a paper strip (15 × 20 cm), folded in zig-zag fashion, as an oviposition substrate. Behavior observations were made during the scotophase [10 h; illumination during the scotophase was provided by a 15 W red light tube (Model No.: 25PCS; Kuanyuan Lighting Co., Ltd., Shantou, China)] because all reproductive behavior occurred during the night for this noctuid moth. Activity of both sexes (Video S1) was observed every 10 min by quickly scanning all pairs by naked eyes and recording via four clear-cut categories: (1) female calling (the female protruding her abdomen between the wings, with the tip bearing the pheromone gland extruded and exposed to the outside [46]); (2) male courtship (the male fanning his wings around the female or exposing his genitalia, trying to engage the female’s genitalia); (3) mating (the two insects engaged by the tip of the abdomen); (4) oviposition (the female protruding her ovipositor to find oviposition site or to lay eggs). The accumulated percentage of mated males or females, and the hourly percentage of courting males, calling females and ovipositing females, before and after mating, were calculated and reported.

### 2.3. Oviposition Pattern

After the two nights’ behavioral observation, as above, females from singly caged virgin female, paired, change male mate and one-time mated female were caged individually, for their lifetime, to observe daily oviposition patterns and lifetime fecundity. Each female was used as a replicate; thus, 20 replicates were used for each treatment (n = 20). Laid eggs were collected daily and incubated in petri dishes (8.5 × 1.5 cm). Egg hatching was recorded 4 days after incubation. Mating was verified by dissecting and counting the number of spermatophores in bursa copulatrix under a dissecting microscope (Olympus SZ61, 6.7–45× total zoom magnification; Olympus Corp., Tokyo, Japan). Differences in lifetime and daily fecundity, including egg hatching rate between treatments, were analyzed using an ANOVA, followed by Tukey’s studentized range (HSD) test for multiple comparison. The percentage data were arcsine transformed prior to analysis. Rejection level was set when α < 0.05 in all analyses. All values are reported as the mean ± SE.

### 2.4. Mating-Induced Transcriptional Changes

#### 2.4.1. Mating Treatments and Sampling

Three-day-old virgin moths were paired immediately after lights off (at the beginning of the scotophase) for mating, and those males and females that started to mate within 2 h after pairing were collected after mating for subsequent RNAseq. The whole body of mated males and females were sampled immediately after mating (0 h after mating) or reared for 6 h or 24 h after mating and then sampled, respectively. Four males or females were combined to form a sample replicate, and three replicates were used for each sampling time point of each sex. Mating was verified by dissecting females to check for the presence of spermatophores (removed from the samples) in the mating sac, as above. The whole body of virgin males and females with the same age as mated ones were used as controls. Immediately after sampling, all samples were frozen in liquid nitrogen and stored at −80 °C.

#### 2.4.2. cDNA Library Preparation and Sequencing

Total RNA was extracted from samples using TRIzol reagent (Invitrogen Inc., Calsbad, CA, USA), according to the manufacturer’s instructions. RNA concentration and purity were assayed by using the Qubit RNA Assay Kit (Life Technologies Inc., Frederick, MD, USA) and a spectrophotometer (Implen Inc., Westlake Village, CA, USA). RNA integrity was measured by using an RNA Nano 6000 Assay Kit (Agilent Technologies Inc., Palo Alto, CA, USA). Sequencing libraries were prepared using the NEBnext Ultra RNA Library Prep Kit for Illumina (New England BioLabs Inc., San Diego, CA, USA), and index codes were added to attribute sequences to each sample. The prepared libraries were sequenced on the Illumina HiSeq 4000 platform (Illumina Inc., San Diego, CA, USA) to generate 125 bp/150 bp paired end reads.

The obtained raw reads were processed to obtain high-quality clean reads by trimming the adapter and reads containing N (N meaning that the base information cannot be determined), and deleting low-quality reads (the number of bases with Qphred ≤ 20 accounts for more than 50% of the total read length) from the raw reads using fastp software (v0.19.5). At the same time, the Q20 (representing an error rate of 1%, meaning every 100 bp sequencing read may contain an error), Q30 (representing an error rate of 0.1%) and GC contents of the clean data were calculated. The clean reads were then mapped to the reference genome sequence of *S. frugiperda* (assembly AGI-APGP CSIRO Sfru_2.0) [47] using Hisat2 software (v2.1.0).

#### 2.4.3. Differential Expression Analysis and Functional Annotation

Gene expression levels were determined by using the transcripts per million (TPM) method. The differential expression analysis between samples was performed using the edgeR R package (v3.24.3). The *p* value was adjusted using the *q*-value [48], and *q* < 0.05 and |log2(foldchange)| > 1 were set as the thresholds for significant differential expression.

The biological function of differentially expressed genes (DEGs) were annotated by comparing DEGs with databases using BLAST software (v2.9.0) [49] by setting the E-value < 1 × 10^−5^ as the threshold. GO enrichment analysis of DEGs was performed using the GOSeq program (v2.12), and KEGG enrichment was performed using KOBAS software (v2.1.1). GO terms and KEGG pathways with *q* < 0.05 were significantly enriched in DEGs.

#### 2.4.4. Validation by qPCR

A total of 14 DEGs were selected, and their expression levels were verified by qPCR, which contained 4 important reproduction related genes (LOC118281239, *microvitellogenin*; LOC118279561 and LOC118265448, *chorion proteins*; LOC118276400, *juvenile hormone-binding protein*), and 2 immunity related genes (LOC118265277, *lysozyme*; LOC118270300, *attacin*), as the main topic of this study, are reproduction and soma maintenance. Total RNA was extracted from samples using RNAiso plus (TaKaRa Inc., Dalian, China), and cDNA was prepared using the PrimeScript RT reagent Kit (Takara Inc., Dalian, China). Real-time quantitative PCR (qPCR) was performed with QuantStudio 7 Flex System (Thermo Fisher Scientific Inc., Carlsbad, CA, USA), using gene-specific primers (Appendix A) and the following program: 95 °C for 30 s, followed by 40 cycles of 95 °C for 5 s, 60 °C for 30 s and dissociation. *Rpl27* (*Ribosomal protein L27*; GenBank ID: XP_035440007.1), which is a housekeeping gene and has often been used as a reference gene in insects, was used as a reference gene. Moreover, *Rpl27* did not show significant transcriptional changes between samples in this study. The 2^−ΔΔCT^ method [50] was used to calculate the relative expression. Differences in gene expression between samples were analyzed using ANOVA by setting the rejection level at *α* < 0.05. All values are reported as the mean ± SE.

## 3. Results

### 3.1. Reproductive Behavior before and after Mating

In the first night, virgin (without mating or before a mating in this night) females and males showed high calling or courting behavior (Figure 1a,c,e,g,i), whereas mated females and males showed very low calling or courting behavior (Figure 1c,e,g,i). The mating rate of paired females (treatment paired, change male mate, change female mate, and one-time mated females) ranged from 57.5 to 65.0%. Calling, courting and mating behavior mostly happened in the earlier part of the night. No females or males mated more than one time and no oviposition happened in this night.

In the second night, most (65–80%) previously mated females (females of paired, change male mate and one-time mated females, mated in the first night) started to lay eggs at the beginning of the night (Figure 2a). Oviposition mostly happened in the earlier part of the night. Fewer (30–40%) of the previously mated females (females of paired and change male mate) mated again in this night, with previously mated or virgin males (Figure 1d,f), whereas more (65%) previously mated males (change female mate) mated again with virgin females in this night (Figure 1h). Similarly, in this night, virgin (Figure 1b) or unmated females and males (without mating or before a mating in this night) (Figure 1d,f,h,j) showed high calling or courting behavior, whereas mated females and males (mated in this night) (Figure 1d,f,h) showed very low calling or courting behavior. However, calling and mating of females from paired and change male mate mostly happened in the later part of the night. No oviposition was observed in singly caged females and females (female 2#) of change female mate in this night.

### 3.2. Oviposition Pattern and Fecundity under Different Mating Conditions

All mated females laid some eggs within three days after the first mating (Figure 2b). A significant difference was found between treatments (paired, change male mate and one-time mated females) on the daily number of eggs laid (1st day, *F*_2,57_ = 3.92, *p* = 0.025; 2nd day, *F*_2,57_ = 5.62, *p* = 0.006; 3rd day, *F*_2,57_ = 3.84, *p* = 0.027; Figure 2c) and larvae (1st day, *F*_2,57_ = 3.29, *p* = 0.044; 2nd day, *F*_2,57_ = 6.02, *p* = 0.004; 3rd day, *F*_2,57_ = 5.05, *p* = 0.01; Figure 2d) in the first three oviposition days. Post hoc pairwise test indicated that one-time mated females have significantly higher eggs or larvae than change male mate in the 1st day (*p* < 0.05), whereas one-time mated female and paired have significantly fewer eggs or larvae than change male mate in the 2nd day (*p* < 0.05), and one-time mated females still have significantly fewer eggs or larvae than change male mate in the 3rd day (*p* < 0.05). No significant difference was found between treatments on the daily number of eggs laid (4th day, *F*_2,57_ = 0.84, *p* = 0.437; 5th day to death, *F*_2,57_ = 0.59, *p* = 0.556; Figure 2c) and larvae (4th day, *F*_2,57_ = 0.54, *p* = 0.585; 5th day to death, *F*_2,57_ = 0.47, *p* = 0.629; Figure 2d) in the 4th day and 5th day to death. Additionally, no significant difference was found between the three treatments on lifetime total eggs laid (*F*_2,57_ = 0.09, *p* = 0.918; Figure 2e), larvae (*F*_2,57_ = 0.26, *p* = 0.768; Figure 2e) and egg hatching rate (*F*_2,57_ = 0.96, *p* = 0.389; Figure 2f).

### 3.3. Mating-Induced Transcriptional Changes

#### 3.3.1. Sequencing Quality

RNAseq obtained ~59,000,000 clean reads from each of the 36 sequenced libraries (Appendix A), with Q20 and Q30 being 97.18–98.15% and 92.14–94.38%, respectively. A total of ~48,000,000 clean reads from each of the libraries were mapped to the genome of *S. frugiperda*, with the mapped ratios ranging from 75.34 to 82.41%. Principal component analysis (PCA) (Appendix A) and Pearson’s correlation coefficient (Appendix A) confirmed the reproducibility of RNASeq and biological replicates. The RNAseq raw reads were deposited into the NCBI SRA database (Accession No.: PRJNA910507).

#### 3.3.2. Summary of Differential Expression Analysis

Mating induced significant transcription changes in *S. frugiperda* females, which showed 352, 1116 and 755 DEGs in mated females (MF) compared to that of virgin females (VF) at 0 h (MF0h-VF0h), 6 h (MF6h-VF6h) and 24 h (MF24h-VF24h) after mating, respectively (Figure 3a–c; Appendix A). Mating also induced significant transcription changes in *S. frugiperda* males, with 1302, 758 and 556 DEGs in mated males (MM) compared to that of virgin males (VM) at 0 h (MM0h-VM0h), 6 h (MM6h-VM6h) and 24 h (MM24h-VM24h) after mating, respectively (Figure 3d–f; Appendix A). In females, there are 34 common DEGs among the three comparison groups (Figure 4a), and in males, there are 38 common DEGs among the 3 comparison groups (Figure 4b). By comparing females with males, 58, 117 and 67 common DEGs were shared by both sexes at 0, 6 and 24 h postmating (Figure 4c–e; Appendix A).

Above DEGs were enriched to GO terms and KEGG pathways (Appendix A). For a better summary and understanding, these enriched terms and pathways were grouped into 13 categories based on their function, including reproduction, soma maintenance (including immunity, stress response, longevity, degradation, apoptosis, detoxication and DNA/protein repair), response (such as response to external biotic stimulus), metabolism, catalysis, transformation, transporter, cell adhesion, cellular component, disease, genetic information, endocrine system and cellular community (Figure 5). Based on these analyses, the important sex-specific mating-responsive molecular changes were explored and described in detail, as follows.

#### 3.3.3. Mating-Induced Transcriptional Changes in Females

At 0 h postmating, there were 166 downregulated DEGs and 186 upregulated DEGs in mated females compared to virgin females (Figure 3a). Downregulated DEGs were enriched to 1 reproduction (steroid hormone biosynthesis), 2 soma maintenance (1 apoptosis and 1 degradation) and 1 metabolism related pathways (Figure 5a; Appendix A). Upregulated DEGs were enriched to 10 reproduction (5 egg development, 3 zygote development and 2 reproductive process), 9 soma maintenance (4 immunity, 3 defense reaction and 2 humoral reaction) and 6 response related terms (Figure 5a; Appendix A).

At 6 h postmating, there were 761 downregulated DEGs and 355 upregulated DEGs (Figure 3b). Downregulated DEGs were enriched to 14 reproductions (6 egg development, 5 zygote development, 2 reproductive processes and 1 steroid hormone biosynthesis), 3 soma maintenance (3 degradation), 2 disease, 46 metabolism, 10 catalysis, 3 transformation and 1 endocrine system related terms/pathways (Figure 5b; Appendix A). Upregulated DEGs were enriched to 1 reproduction (insect hormone biosynthesis), 13 soma maintenance (5 immunity, 3 defense reaction, 2 humoral reaction and 3 degradation), 6 response, 11 metabolism, 6 catalysis, 2 transformation, 2 transporter, 3 cell adhesion and 2 genetic information related terms/pathways (Figure 5b; Appendix A).

At 24 h postmating, there were 454 downregulated DEGs and 301 upregulated DEGs (Figure 3c). Downregulated DEGs were enriched to 6 metabolism, 2 catalysis and 2 transformation related terms/pathways (Figure 5c; Appendix A). Upregulated DEGs were enriched to 16 reproduction (6 egg development, 5 zygote development, 2 reproductive process, 2 reproduction related hormone and 1 estrogen signaling), 12 maintenance (4 immunity, 3 defense reaction, 2 humoral reaction, 1 antigen processing, 1 apoptosis and 1 longevity), 6 response and 5 disease related terms/pathways (Figure 5c; Appendix A).

#### 3.3.4. Mating-Induced Transcriptional Changes in Males

At 0 h postmating, there were 813 downregulated DEGs and 489 upregulated DEGs in mated males compared to virgin males (Figure 3d). Downregulated DEGs were enriched to 1 soma maintenance (double-strand break repair), 13 metabolism, 3 catalysis, 12 transformation, 5 cellular component and 1 genetic information related terms/pathways (Figure 5d; Appendix A). Upregulated DEGs were enriched to 13 soma maintenance (5 immunity, 3 defense reaction, 2 humoral reaction and 3 degradation), 6 response, 4 metabolism, 2 transformation and 1 disease related terms/pathways (Figure 5d; Appendix A).

At 6 h postmating, there were 423 downregulated DEGs and 335 upregulated DEGs (Figure 3e). Downregulated DEGs were enriched to 9 soma maintenance (4 immunity, 3 defense reaction and 2 degradation), 6 response, 3 metabolism, 2 catalysis, 7 transformation and 1 genetic information related terms/pathways (Figure 5e; Appendix A). Upregulated DEGs were enriched to 2 metabolism, 8 catalysis and 1 genetic information related terms/pathways (Figure 5e; Appendix A).

At 24 h postmating, there were 256 downregulated DEGs and 300 upregulated DEGs (Figure 3f). Downregulated DEGs were enriched to 13 maintenance (5 immunity, 3 defense reaction, 2 humoral reaction, 2 cytolysis and 1 phagosome), 8 response, 1 catalysis, 7 transformation, 7 cellular component, 3 disease and 1 cellular community related terms/pathways (Figure 5f; Appendix A). Upregulated DEGs were not enriched to any terms/pathways.

Male DEGs were not enriched to any reproduction related terms/pathways.

#### 3.3.5. Mating-Induced Common Changes between Females and Males

A total of 228 common DEGs were identified in mating-induced changes between females and males (Figure 4c–e; Appendix A). These DEGs were enriched to 13 soma maintenance, 8 response, 13 metabolism, 2 catalysis, 4 transformation terms, 2 metabolism and 1 genetic information pathways (Appendix A). None of these DEGs were enriched to any reproduction related terms/pathways.

#### 3.3.6. Validation of RNAseq Results by qPCR

The results showed that the expression levels of target genes (Figure 6) were similar to the results from the RNAseq analysis, suggesting that the RNAseq data were reliable.

## 4. Discussion

Behavioral observation indicated that mating triggered remarkable behavioral changes in both sexes of *S. frugiperda*, particularly in females (Figure 1). Virgin females usually like to conduct calling behavior, whereas mated females rarely engaged in calling behavior and did not easily accept male mating trials. Similar behavioral switches before and after mating were also found in many other insects, such as *D. melanogaster* [51] from Diptera, and *S. litura* [31], *E*. *kuehniella* [52] and *H. armigera* [29] from Lepidoptera. In the present study, we noticed that mated females did show very low calling behavior in the time of night that was left after mating (Figure 1c,e,i), while they conducted calling behavior normally in the next night before mating again (Figure 1d,f,j). Most noctuid female moths produce species-specific sex pheromones, which are released during calling to attract males [53]. A previous study in *H. zea* found that MAG produce a pheromonostatic peptide (PSP) that has a short-term (few hours) effect in depleting pheromones of females, thus making them unreceptive to males [54]. A later study in *S. litura* also demonstrated that seminal fluid extract injection can suppress female calling behavior for a few hours [27]. These may explain the temporary suppression of calling behavior in mated *S. frugiperda* females. This study also found that mating induced a temporary inhibition of courting behavior in *S. frugiperda* males. Studies in a number of moths also showed that mating induced transient sexual inhibition or abstinence in males [55,56]. In the noctuid moths of *Agrotis ipsilon* [55] and *S. littoralis* [57], mating in males temporarily abolished neuronal and olfactory responses to the female sex pheromone. A recent study in *A. ipsilon* further revealed that *Neuroligin 1* expression is linked to plasticity of behavioral and neuronal responses to the sex pheromone in males [58].

The results also indicated that both male and female *S. frugiperda* did not copulate more than once during the same scotophase. Previous studies in other lepidopterans also found that both males and females seldom copulated more than once during the same scotophase, such as *A. ipsilon* [55] and *S. litura* [59]. This may also mainly be due to the above mentioned mating-induced suppression of sexual receptivity in both sexes. A refractory period, or the transient inhibition of sexual behavior, allows males to replenish sperm and accessory gland secretions [28], and helps both sexes to balance energy metabolism and avoiding fruitless arousal and sexual activity [28,55,57]. The present study also showed that females did not lay eggs until the next day after the first mating. This may be because female moths usually have a long ductus seminalis (the duct connecting between the bursa copulatrix and spermatheca), and sperm migration from the spermatophore to the spermatheca, and sperm maturation in spermatheca, usually needs quite a few hours [60,61]. A long ductus seminalis may be evolved to promote sperm competition by favoring the ‘vigorous’ sperm that could reach the spermatheca and fertilize eggs [62].

In many insect species, one mating is adequate for a female to obtain her maximum reproductive output, whereas the majority of females prefer to mate multiply, with the same or different males [63]. Females may obtain a higher fertilization rate or more offspring, due to sperm replenishment, and more nutritious ejaculates from multiple mating [63,64,65]. However, the fecundity and egg hatching test in *S. frugiperda* did not find significant differences between treatments on lifetime reproductive output (Figure 2c,d) and the egg hatching rate (Figure 2f). Moreover, daily fecundity analysis revealed significant differences between treatments on the number of eggs laid and larvae in the first three oviposition days, whereas females of one-time mated female have significantly higher eggs or larvae than females of change male mate in the 1st day, and females of one-time mated female and paired have significantly lower eggs or larvae than females of change male mate in the 2nd day; one-time mated females also have significantly lower eggs or larvae than females of change male mate in the 3rd day (Figure 2c,d). Similar oviposition patterns under different mating conditions have also been noticed in a number of insect species, including *Dermestes maculatus* [66] from Coleoptera, and *E. kuehniella* [67] and *S. litura* [68] from Lepidoptera. These results suggest that females may intentionally retain eggs to be fertilized by novel males or to be fertilized competitively by different males, which maximizes the genetic incompatibility benefits of polyandry [69,70,71,72] and genetic diversity in offspring [73,74].

RNAseq and differential expression analysis found that mating also induced significant transcriptional changes in both females and males of *S. frugiperda* (Figure 3). In females, the transcriptional changes were relatively lower immediately after mating (0 h postmating), peaked at 6 h postmating and then declined at 24 h postmating; in males, the transcriptional changes were high immediately after mating and then declined with time.

Previous studies generally found that mating induces the upregulation of egg production and fertilization rate [32,35,36,41], whereas in *C. chuxiongica,* mating induced divergent reproductive response, with downregulation on genes related to egg production, while upregulation on genes related to egg fertilization [38]. In the present study, functional enrichment analysis in *S. frugiperda* females showed there is more reproduction related terms/pathways enriched to upregulated DEGs than to downregulated DEGs (10:1) at 0 h postmating, which is less (1:14) at 6 h postmating, and then more (16:0) again at 24 h postmating. These terms/pathways mainly relate to egg development, zygote development and reproductive processes. This result suggests that mating induced large scale transcriptional regulation on genes and functions related to reproduction, and the change pattern should be consistent with postmating physiological and behavioral changes, i.e., high physiological and behavioral intensity during mating (0 h postmating), lower behavioral activities (such as very low calling behavior) a few hours after mating (6 h postmating) (Figure 1c,e,i) and high reproductive activities at 24 h postmating (females start to lay a large number of eggs) (Figure 2a). However, in *S. frugiperda* males, the mating-induced DEGs did not enrich to any reproduction related terms or pathways. This may be because male reproductive bioinformatics is relatively limited. For example, there are quite a number of Acps that have been identified in *Drosophila* [20], whereas evidence on Acps in moths is still scarce. This may be because Acps evolve rapidly and have a large interspecific difference due to the targeted selection of intrasexual competition and sexual conflict [75,76]. No Acps were annotated in the present study or in the genome sequencing of *S. frugiperda* [47].

Previous studies have shown that mating may also promote trade-offs between reproduction and survival, with increased reproductive activities after mating being likely to incur cost on soma maintenance [36,37,38,41]. In the present study, however, mating induced upregulation on response (such as response to external biotic stimulus, response to bacterium and response to external stimulus; Appendix A) and soma maintenance (such as immune activity, stress reaction and detoxification; Appendix A) related processes in *S. frugiperda* females at 0, 6 and 24 h postmating (Figure 5a–c). In males, mating also induced upregulation on response and soma maintenance related processes at 0 h postmating, but induced downregulation on these and most other processes at 6 and 24 h postmating (Figure 5d–f). This regulation pattern may also be related to sex-specific physiological and behavioral changes after mating (Figure 1 and Figure 2), and the difference on reproductive systems between males and females, but may not be related to trade-offs between reproduction and survival. Mating can challenge the female’s immune system by transferring foreign materials and infections to females [77,78,79,80]; females of polyandrous species should have higher postmating immunity if sexually transmitted infection is the major factor driving female postmating costs [80]. Therefore, mating-induced transcriptional changes should be sex-specific (this study) and species- and tissue-specific [36,41].

In conclusion, mating induced common and sex-specific behavioral changes in *S. frugiperda*. Mating also induced large scale transcriptional changes on genes related to reproduction and soma maintenance in *S. frugiperda* males and females, and the change pattern is related to sex-specific postmating behavioral changes and the difference on reproductive systems between males and females.

## Figures and Tables

**Figure 1 insects-14-00209-f001:**
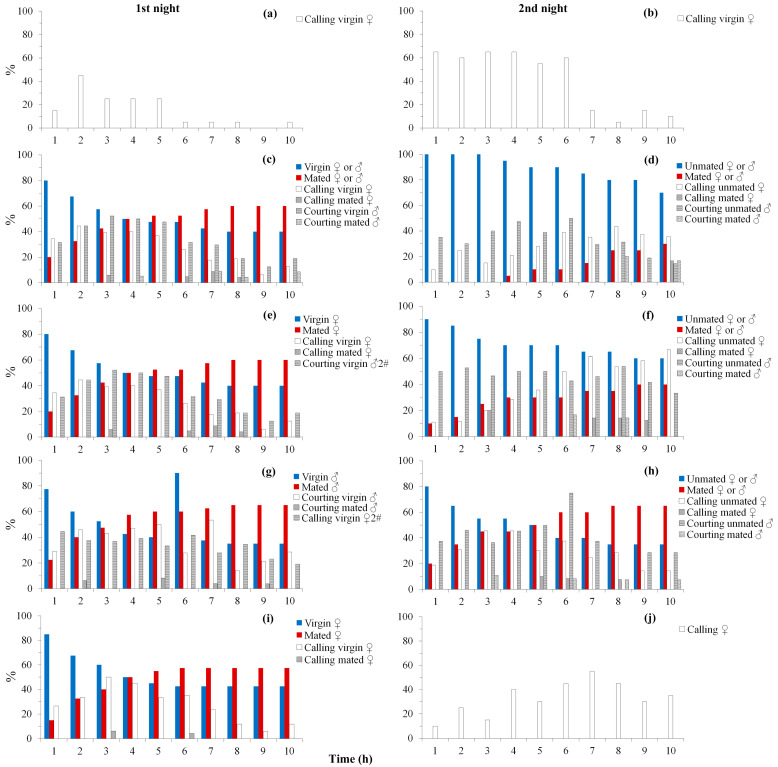
Reproductive behavior before and after mating, under different mating conditions, in *S. frugiperda*. (**a**,**b**) are behaviors of singly caged virgin females; (**c**,**d**) are behaviors of females and males from paired; (**e**,**f**) are behaviors of females and males (male 2#) from change male mate; (**g**,**h**) are behaviors of males and females (female 2#) from change female mate; (**i**,**j**) are behaviors of one-time mated females. The observation was performed for two nights. In each night, the percentages of mated males or females are accumulated values, and the percentage of virgin or unmated males or females = 100% − the percentage of mated males or females, whereas the percentages of courting males, calling females and ovipositing females are the exact values of an hour, such that the percentage of ovipositing females of a given hour = the number of ovipositing females observed in this hour/the number of total females %.

**Figure 2 insects-14-00209-f002:**
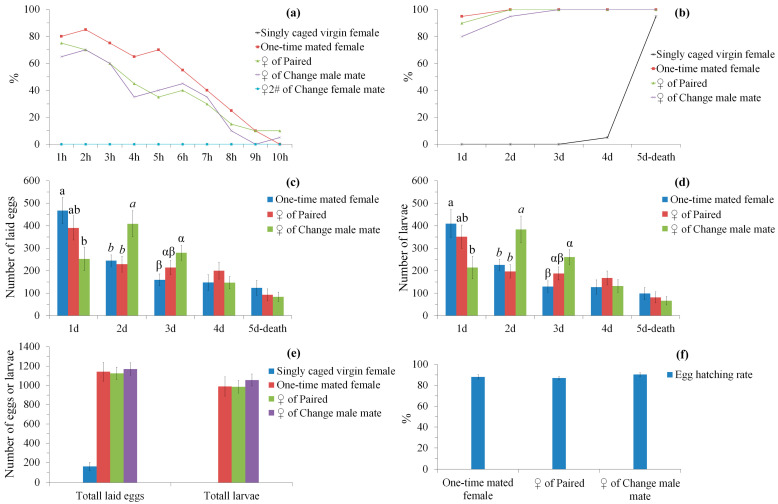
Oviposition pattern and fecundity under different mating conditions in *S. frugiperda* (20 females were used for each treatment). (**a**) hourly percentage of ovipositing females. The percentage of an hour in a treatment = the number of ovipositing females observed in this hour/the number of total females %; (**b**) accumulated percentage of females that have laid eggs; (**c**) daily fecundity (number of eggs laid) per female; (**d**) daily hatched eggs (number of larvae) per female; (**e**) lifetime total laid eggs and larvae per female; (**f**) egg hatching rate per female. In each oviposition day of subgraph (**c**,**d**), bars with different letters are significantly different (*p* < 0.05).

**Figure 3 insects-14-00209-f003:**
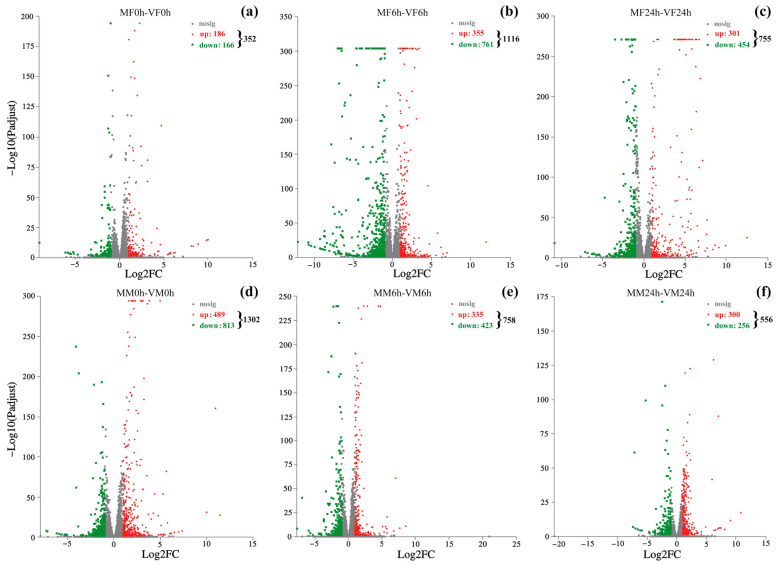
Volcano plots of DEGs in *S. frugiperda*. (**a**–**c**) are DEGs of mated vs. virgin females at 0, 6 and 24 h postmating, respectively; (**d**–**f**) are DEGs of mated vs. virgin males at 0, 6 and 24 h postmating, respectively. Genes with significantly differential expression are indicated by red dots (upregulated) and green dots (downregulated). Blue dots indicate genes with no significant differential expression.

**Figure 4 insects-14-00209-f004:**
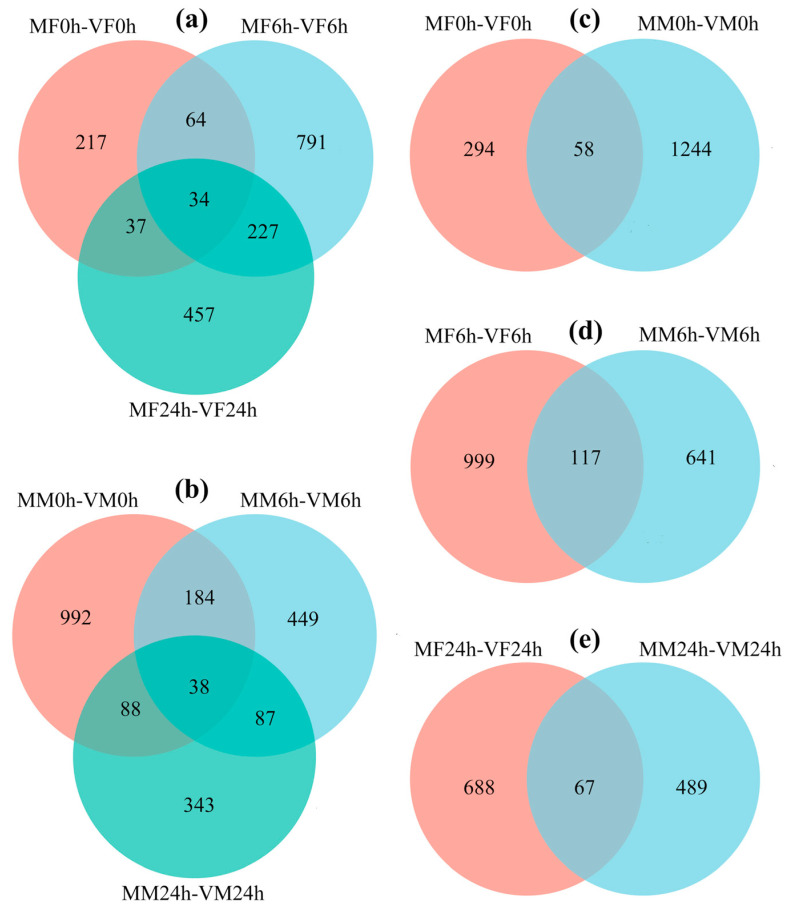
Venn diagrams of DEGs in *S. frugiperda*. (**a**,**b**) are common DEGs in females and males at 0, 6 and 24 h postmating, respectively; (**c**–**e**) are common DEGs shared by both sexes at 0, 6 and 24 h postmating, respectively. The overlapping circles represent common DEGs among all combinations.

**Figure 5 insects-14-00209-f005:**
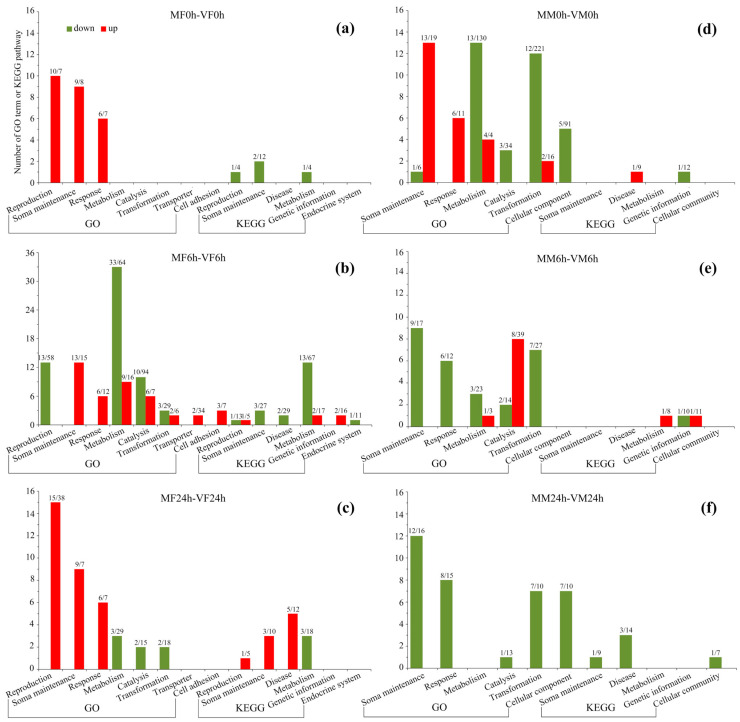
Enriched GO terms and KEGG pathways in *S. frugiperda*. (**a**–**c**) represent terms and pathways in females at 0, 6 and 24 h postmating, respectively; (**d**–**f**) represent terms and pathways in males at 0, 6 and 24 h postmating, respectively. Red columns represent terms/pathways enriched to upregulated DEGs and green columns represent terms/pathways enriched to downregulated DEGs. At the top of columns, the first number represents the number of terms/pathways and the second number represents the number of enriched DEGs.

**Figure 6 insects-14-00209-f006:**
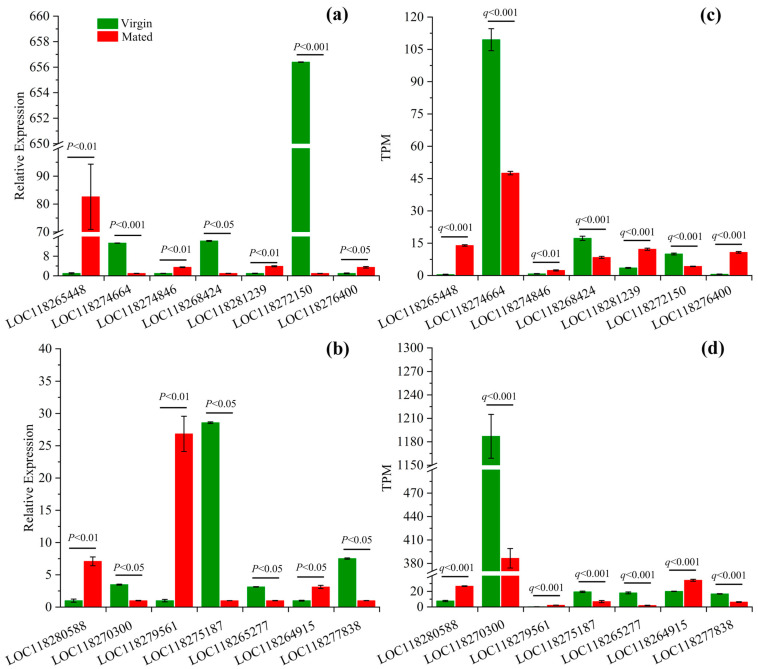
Transcriptome validation for DEGs by qPCR. (**a**) relative expression levels of target genes in females measured by qPCR and (**c**) their expression levels measured by RNAseq; (**b**) relative expression levels of target genes in males measured by qPCR and (**d**) their expression levels measured by RNAseq.

## Data Availability

The transcriptome raw reads have been deposited to the NCBI SRA database; the accession number is PRJNA910507. Other data generated or analyzed during this study are included in this article and its Appendix A.

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
