# Peer review of "Mating-Induced Common and Sex-Specific Behavioral, Transcriptional Changes in the Moth Fall Armyworm (Spodoptera frugiperda, Noctuidae, Lepidoptera) in Laboratory"

_insects, 2023, doi:10.3390/insects14020209_

Round 1

Reviewer 1 Report

In reproductive and evolutionary biology, there are concepts of pre- and postcopulative isolation, as well as pre- and postzygtic isolation. Although the terms “precopulative” and “prezygotic” are close in the meaning and refer to very close chronological moments in the life cycle of an organism, the distinction between them is very important, since each of them characterizes a specific critical moment. It should also be emphasized that the period between copulation and the appearance of the zygote is a kind of gray area in biology, very poorly understood both from the point of view of behavioral and physiological changes.

The authors of the article invade this gray area, which is certainly important and should arouse the interest of readers. In addition, a significant positive aspect of the work is that the authors combine the classical approach based on the study of behavior with transcriptomics. Therefore, I positively evaluate the manuscript, as it is a significant contribution to the knowledge of general issues of reproductive biology, and particular biology of the moth Spodoptera frugiperda.

Having said this, I want to note that the work, in my opinion, is not clearly integrated into the system of already existing knowledge regarding the processes being studied. The authors write that they “studied the mating-induced common and sex-specific transcriptional and behavioral changes in S. frugiperda” and that they “also discussed the evolutionary significance of the postmating gene expression regulation between reproduction  and soma maintenance and the molecular machinery behind the postmating switches”. This is great, but it would be useful to formulate a specific hypothesis that the authors are testing, and in the discussion not only state the technical points, but also show how the data obtained led to the confirmation, rejection or modification of this hypothesis. It would also be useful to show what the specific scientific novelty of the work is.

A more specific note: abbreviations appear already in the abstract, but they are not deciphered there.

The authors provide a large list of additional (supplementary) materials. Nevertheless, it would be great to see one more supporting material: documentation of behavior before and after copulation. It's not obligatory, but a short video about it would greatly enhance the work.

Author Response

In reproductive and evolutionary biology, there are concepts of pre- and postcopulative isolation, as well as pre- and postzygtic isolation. Although the terms “precopulative” and “prezygotic” are close in the meaning and refer to very close chronological moments in the life cycle of an organism, the distinction between them is very important, since each of them characterizes a specific critical moment. It should also be emphasized that the period between copulation and the appearance of the zygote is a kind of gray area in biology, very poorly understood both from the point of view of behavioral and physiological changes.

The authors of the article invade this gray area, which is certainly important and should arouse the interest of readers. In addition, a significant positive aspect of the work is that the authors combine the classical approach based on the study of behavior with transcriptomics. Therefore, I positively evaluate the manuscript, as it is a significant contribution to the knowledge of general issues of reproductive biology, and particular biology of the moth Spodoptera frugiperda.

Our answer: We appreciate the positive comments to this MS.

Having said this, I want to note that the work, in my opinion, is not clearly integrated into the system of already existing knowledge regarding the processes being studied. The authors write that they “studied the mating-induced common and sex-specific transcriptional and behavioral changes in S. frugiperda” and that they “also discussed the evolutionary significance of the postmating gene expression regulation between reproduction  and soma maintenance and the molecular machinery behind the postmating switches”. This is great, but it would be useful to formulate a specific hypothesis that the authors are testing, and in the discussion not only state the technical points, but also show how the data obtained led to the confirmation, rejection or modification of this hypothesis. It would also be useful to show what the specific scientific novelty of the work is.

Our answer: We agree and have now revised the paper very carefully according to these and similar constructive comments from other reviewers. We have now clearly developed a number of the hypotheses (now see the last paragraph of the Introduction) and tested them by using a series of experiments (see Results) and then obtained the confirmation or rejection based on the results and discussion (Mainly see the last two paragraph of Discussion). We also discussed other important findings thoroughly upon the view of Lepidoptera, such as on behaviour and oviposition patterns.

A more specific note: abbreviations appear already in the abstract, but they are not deciphered there.

Our answer: Full words were included now.

The authors provide a large list of additional (supplementary) materials. Nevertheless, it would be great to see one more supporting material: documentation of behavior before and after copulation. It's not obligatory, but a short video about it would greatly enhance the work.

Our answer: We did not use the camera for behavioural recording due to the light is weak during the night and there are too many replicates, but through the naked eyes for behavioural observation. However, we do have a short video (but not very clear due the weak light during the night) that recorded the reproductive behaviour of paired S. frugiperda and have now uploaded to supplementary materials.

Reviewer 2 Report

This is a technical paper on the reproductive behaviour of the noctuid moth Spodoptera frugiperda, introduced to China, where it appears as a serious pest. Therefore it is important to know as much biological details of the species as possible for a successful crop protection; so the objective of the paper is absolutely well grounded. The experiment is well designed and the work is precisely carried out. However in the present form I do not recommend the manuscript for publication. It needs a major revision. My points are listed below.

1. The paper does not mention any of the experimental works carried in laboratory on noctuid moths, like Agrotis exclamationis, A. ipsilon, A. segetum, Autographa gamma, Helicoverpa armigera, etc.  and published in numerous papers. These closely related organisms represent the order Lepidoptera and family Noctuidae, like the subject of the paper, hence they are more appropriate to any kind of comparision than Bemesia (Hemiptera), Cephalica (Hymenoptera), and Drosophila (Diptera). At least an additional paragraph should be dedicated to the experiments carried out on these moths in the introduction, and also it would be important to take them as comparative subjects in the discussion.

2. These experimental papers must be taken into consideration whether the behavioral tarits examined by the paper were in their subjects or not. If not that must be pointed out why these aspects are important for a better understanding of the species.

3.  There are given a lot of technical details, they are difficult to decipher or overview. For example the description of the five kinds of  experiments (lines 132-153) plus an additional one is almost impossible to follow; it is crucial to compose these technical details into tables, or /and refrase the sentences.

3. In contrary, there is no technical details how the observations were recorded (scanned) under 15W red light tube. It is crucial to indicate the spectum of the illumination to ensure that the experiments can be repeated. So please give the details of the light tube, the details of the camera; the artificial food, the origin of the moth stock,  etc. etc.

4.. Miscellanea

- title: there are two "the"s; the order and family indications have to be given, and probably it would be necessary somewhow to indicate that the experiments were carried out in laboratory; probably like this: "Mating-induced common and sex-specific behavioral, and transcriptional changes in the moth Fall Armyworm (Spodoptera frugiperda, Noctuidae, Lepidoptera) in laboratory"

- lines 50-51: "This moth pest was first found"; I guess that the moth was found first, and later it was recorded also as pest in China; the caterpillars attack the corn (that is the source of the the common English name)

- line 26: the reference [39] is strange, a proper reference or the recept of the artificial diet should be given

- line 393: "Virigin females rarely lay eggs"; but when they do, why? these eggs are not fertilized (see here: doi: 10.1673/031.009.5001); probably this semisentence should be delated becaesu it is confusing here

Author Response

This is a technical paper on the reproductive behaviour of the noctuid moth Spodoptera frugiperda, introduced to China, where it appears as a serious pest. Therefore it is important to know as much biological details of the species as possible for a successful crop protection; so the objective of the paper is absolutely well grounded. The experiment is well designed and the work is precisely carried out. However in the present form I do not recommend the manuscript for publication. It needs a major revision. My points are listed below.

Our answer: We thank the positive comments and revision suggestions to our MS.

  1. The paper does not mention any of the experimental works carried in laboratory on noctuid moths, like Agrotis exclamationis, A. ipsilon, A. segetum, Autographa gamma, Helicoverpa armigera, etc.  and published in numerous papers. These closely related organisms represent the order Lepidoptera and family Noctuidae, like the subject of the paper, hence they are more appropriate to any kind of comparision than Bemesia (Hemiptera), Cephalica (Hymenoptera), and Drosophila (Diptera). At least an additional paragraph should be dedicated to the experiments carried out on these moths in the introduction, and also it would be important to take them as comparative subjects in the discussion.

Our answer: We appreciate this constructive comment and have now summarised the related studies on Lepidoptera (now see the 4th paragraph of Introduction) and have now discussed the important findings accordingly, mainly from the perspective of lepidopterans.

  1. These experimental papers must be taken into consideration whether the behavioral tarits examined by the paper were in their subjects or not. If not that must be pointed out why these aspects are important for a better understanding of the species.

Our answer: Agree and please see our answer to the above point (point 1).

  1. There are given a lot of technical details, they are difficult to decipher or overview. For example the description of the five kinds of  experiments (lines 132-153) plus an additional one is almost impossible to follow; it is crucial to compose these technical details into tables, or /and refrase the sentences.

Our answer: We agree and have now revised this and other parts of the Method for easy understanding. It only has five treatments (no an additional one).

  1. In contrary, there is no technical details how the observations were recorded (scanned) under 15W red light tube. It is crucial to indicate the spectum of the illumination to ensure that the experiments can be repeated. So please give the details of the light tube, the details of the camera; the artificial food, the origin of the moth stock,  etc. etc.

Our answer: We have now given the details of the light tube, the recipe of the food, the moth stock and other necessary issues. We did not use the camera for behavioural recording due to the light is weak during the night and there are too many replicates, but through the naked eyes for behavioural observation. However, we do have a short video that recorded the reproductive behaviour of paired S. frugiperda and have now uploaded to supplementary materials as suggested by reviewer 1#.

4.. Miscellanea

- title: there are two "the"s; the order and family indications have to be given, and probably it would be necessary somewhow to indicate that the experiments were carried out in laboratory; probably like this: "Mating-induced common and sex-specific behavioral, and transcriptional changes in the moth Fall Armyworm (Spodoptera frugiperda, Noctuidae, Lepidoptera) in laboratory"

Our answer: Thanks and revised the title accordingly.

- lines 50-51: "This moth pest was first found"; I guess that the moth was found first, and later it was recorded also as pest in China; the caterpillars attack the corn (that is the source of the the common English name)

Our answer: Agree and revised this sentence.

- line 26: the reference [39] is strange, a proper reference or the recept of the artificial diet should be given

Our answer: A proper reference was used and detailed information on diet preparation has been given now.

- line 393: "Virigin females rarely lay eggs"; but when they do, why? these eggs are not fertilized (see here: doi: 10.1673/031.009.5001); probably this semisentence should be delated becaesu it is confusing here

Our answer: Agree and revised this sentence.

Reviewer 3 Report

The manuscript needs some  improvements, the suggestions are as followings:
Abstract
-Please add a sentence of the objective of the research after the sentence of the background of the 1st sentence.

-Please add a sentence for the conclusion before the sentence “These results suggest that ...”

1. Introduction

-The  Introduction was too long: Please shorten the introducton to be a maximum of 600-700 words consisted three or four paragraphs (containing the research questions/background; state of the art; and gap analysis & novelty). it is unnecessary for describing about the mating factors at the 3rd paragraph in detail (extensively) as well the  postmating changes at the 4th paragraph, the both paragraph should be merged to be a short paragraph.  The 6th and 7th paragraph are unnecessary to be mentioned in this section.

- This pest mainly  attacks corn ...(line 51) and   now has two strains (please cite the article of ... https://www.sciencedirect.com/science/article/pii/S1658077X21001533 )

-Please add the objective of the research at the last paragraph in this section.

2. Materials and methods

2.1. Insects: in this section please mention the laboratory where the research  was carried out with the coordinate point to indicate the position of the lab.

3. Results

Well written

Discussion

-Please add a last paragraph for the conclusion at this section.

Author Response

The manuscript needs some  improvements, the suggestions are as followings:

Our answer: We thank the reviewer for these constructive revision suggestions.

Abstract
-Please add a sentence of the objective of the research after the sentence of the background of the 1st sentence.

-Please add a sentence for the conclusion before the sentence “These results suggest that ...”

Our answer: Thanks and revised the Abstract accordingly.

  1. Introduction

-The  Introduction was too long: Please shorten the introducton to be a maximum of 600-700 words consisted three or four paragraphs (containing the research questions/background; state of the art; and gap analysis & novelty). it is unnecessary for describing about the mating factors at the 3rd paragraph in detail (extensively) as well the  postmating changes at the 4th paragraph, the both paragraph should be merged to be a short paragraph.  The 6th and 7th paragraph are unnecessary to be mentioned in this section.

Our answer: We agree and have now largely shortened the Introduction. We also added some more relevant information (mainly for Lepidoptera) according to comments from other reviewers.

- This pest mainly  attacks corn ...(line 51) and   now has two strains (please cite the article of ... https://www.sciencedirect.com/science/article/pii/S1658077X21001533 )

Our answer: Thanks and included now.

-Please add the objective of the research at the last paragraph in this section.

 Our answer: Done.

  1. Materials and methods

2.1. Insects: in this section please mention the laboratory where the research  was carried out with the coordinate point to indicate the position of the lab.
Our answer: Done.

  1. Results

Well written

Our answer: Thanks for the reviewing.

Discussion

-Please add a last paragraph for the conclusion at this section.

Our answer: Agree and added a paragraph.

Reviewer 4 Report

Dear authors

Thank you for your manuscript that is quite interesting but you should include more details to facilitate the reproduction of your work and its understanding.

Please make a few changes in your simple summary, it is very similar to your abstract.

Introduction

L21 Include the meaning of DEGs

L25 Use a synonym of ‘change’ to avoid two changes

L56 strong drug resistance or insecticide resistance, please double check your sentence

L76 -77 There are some works done in Aedes albopictus and Ae. aegypti. You should read them.

L95 Write RNAseq in full word since it is the first time you mentioned it.

Methods

L126 Include the name and manufacturer of the artificial diet

L168-169 Check the meaning of your sentence

L173-174 Include the total magnification, model/brand of your dissecting microscope

L201-202 Include the software/tool for trimming and do not forget to include its parameters.

L203-205 Remove the link and include the genome version

L207-215 Include the parameters of all the used software/programs such as edgeR, GOSeq program, KOBAS and others.

L220-221 Include the cycle of dissociation

Results

Figure 2 remove ‘is’’ after the letters a, b, c, d, e and f and include the total number of females for each condition. Example, figure 2 c should be the number of laid eggs per female.

L284-285 Define Q20 and Q30

You should include a few sentences to interpret your results : For instance after  L350, L372 and others.

L380 Explain how you select your DEGs for qPCR

L386 Is it possible to include a correlation coefficient between your RNAseq and qPCR ? Or just represent them by using the same graph.

Discussion

L410-412 Check the meaning of your sentence or clarify it.

Author Response

Thank you for your manuscript that is quite interesting but you should include more details to facilitate the reproduction of your work and its understanding.

Our answer: We are very grateful for the constructive comments and suggestions. We have now revised the MS very carefully according to the following comments.

Please make a few changes in your simple summary, it is very similar to your abstract.

Our answer: Agree and revised the simple summary.

Introduction

L21 Include the meaning of DEGs

Our answer: Agree and provided the whole name.

L25 Use a synonym of ‘change’ to avoid two changes

Our answer: Done.

L56 strong drug resistance or insecticide resistance, please double check your sentence

Our answer: Thanks and crrected.

L76 -77 There are some works done in Aedes albopictus and Ae. aegypti. You should read them.

Our answer: Agree and have now provided relevant information on these species (now see the third paragraph of Introduction).

L95 Write RNAseq in full word since it is the first time you mentioned it.

Our answer: Done.

Methods

L126 Include the name and manufacturer of the artificial diet

Our answer: The diet was prepared by the author according to published recipe. Detailed information was provided now.

L168-169 Check the meaning of your sentence

Our answer: Revised.

L173-174 Include the total magnification, model/brand of your dissecting microscope

Our answer: Done.

L201-202 Include the software/tool for trimming and do not forget to include its parameters.

Our answer: Provided now.

L203-205 Remove the link and include the genome version

Our answer: Done.

L207-215 Include the parameters of all the used software/programs such as edgeR, GOSeq program, KOBAS and others.

Our answer:

L220-221 Include the cycle of dissociation

 Our answer: The qPCR cycle (40) has been provided and the dissociation of each cycle were detected.

Results

Figure 2 remove ‘is’’ after the letters a, b, c, d, e and f and include the total number of females for each condition. Example, figure 2 c should be the number of laid eggs per female.

 Our answer: Agree and revised accordingly.

L284-285 Define Q20 and Q30

 Our answer: They have been defined now in the Method part.

You should include a few sentences to interpret your results : For instance after  L350, L372 and others.

 Our answer: We think these results have been fully interpreted in the Discussion part (such as L448-L454).

L380 Explain how you select your DEGs for qPCR

Our answer: Agree and explained now in the Method part.

L386 Is it possible to include a correlation coefficient between your RNAseq and qPCR ? Or just represent them by using the same graph.

Our answer: We suggest keeping the current graph as it is more direct and often used in many publications.

Discussion

L410-412 Check the meaning of your sentence or clarify it.

Our answer: Revised.

Round 2

Reviewer 2 Report

Thank your for your kind work. The version is better, please find below some remarks, what probably could help the paper to be even better. Greetings.

1 - there are small typos, what should be corrected (e.g. in the title after the word "Armyworm" you need a space; genus-group and species-group names should be formatted consistently in italics, etc.)

2 - In my review I suggested that the extensive experimental work on noctuid pests at least should be taken into consideration, briefly reviewed, and if they do not relevant to the subject, that should be mentioned. This would make the paper stronger, as it stresses the novelty of the experiment you carried on Spodoptera frugiperda. You added five new references, but this is not the whole palette.

Using  google search typing the words "female calling behaviour Autographa gamma" I got this as first item: https://www.jstor.org/stable/3548037

Using  google search typing the words "female calling behaviour Agrotis segetum" I got this as first item: https://doi.org/10.1023/B:JOEC.0000006454.84415.5f

Using  google search typing the words "female calling behaviour Agrotis ipsilon" I got this as first item: https://doi.org/10.1093/aesa/70.6.919

etc.

3 - pp. 161-162: "by naked eyes", this would be more comforting to change: "scanning all pairs by by naked eyes recording via four clear-cut categories as: (1) female calling (the female protruding...), (2) male courtship (the male fanning), (3) mating (the two insects...), and (4) oviposition (the female protruding...)".

Author Response

Thank your for your kind work. The version is better, please find below some remarks, what probably could help the paper to be even better. Greetings.

Our answer: Thank you again for the reviewing and further helpful comments.

1 - there are small typos, what should be corrected (e.g. in the title after the word "Armyworm" you need a space; genus-group and species-group names should be formatted consistently in italics, etc.)

Our answer: We agree and have now checked the whole MS carefully and made necessary changes.

2 - In my review I suggested that the extensive experimental work on noctuid pests at least should be taken into consideration, briefly reviewed, and if they do not relevant to the subject, that should be mentioned. This would make the paper stronger, as it stresses the novelty of the experiment you carried on Spodoptera frugiperda. You added five new references, but this is not the whole palette.

Using  google search typing the words "female calling behaviour Autographa gamma" I got this as first item: https://www.jstor.org/stable/3548037

Using  google search typing the words "female calling behaviour Agrotis segetum" I got this as first item: https://doi.org/10.1023/B:JOEC.0000006454.84415.5f

Using  google search typing the words "female calling behaviour Agrotis ipsilon" I got this as first item: https://doi.org/10.1093/aesa/70.6.919

etc.

Our answer: We thank these supporting comments and added 8 relevant lepidopteran references (5 on noctuid moths) in Introduction and Discussion, which has provided sufficient information for both sexes in this field.

3 - pp. 161-162: "by naked eyes", this would be more comforting to change: "scanning all pairs by by naked eyes recording via four clear-cut categories as: (1) female calling (the female protruding...), (2) male courtship (the male fanning), (3) mating (the two insects...), and (4) oviposition (the female protruding...)".

Our answer: Thanks and revised accordingly.

Reviewer 4 Report

The revised version of the manuscript is much better as compared to the original. The Authors should double check English mistakes and other errors before its publication. For instance, Figure 6 is not visible and "is'' should be removed after the letters a, b, c, d.

Author Response

The revised version of the manuscript is much better as compared to the original. The Authors should double check English mistakes and other errors before its publication. For instance, Figure 6 is not visible and "is'' should be removed after the letters a, b, c, d.

Our answer: We thank these further comments and have now checked the whole MS very carefully and made necessary corrections.